# Effects on the Synthesis and Accumulation of Triterpenes in Leaves of *Cyclocarya paliurus* under MeJA Treatment

**Qinghui Xia** [1,2,†], **Zijue Wang** [1,2,†], **Xiaoling Chen** [1,3,*], **Xingxing Dong** [1,3], **Shuiyuan Cheng** [1,3] **and Shaopeng Zhang** [1,3]

1. School of Modern Industry for Selenium Science and Engineering, Wuhan Polytechnic University, Wuhan 430023, China; 15571778723@163.com (Q.X.); 13039522428@163.com (Z.W.); dongxingxinghg@whpu.edu.cn (X.D.)
2. School of Life Science and Technology, Wuhan Polytechnic University, Wuhan 430023, China
3. National R&D Center for Se-Rich Agricultural Products Processing Technology, Wuhan Polytechnic University, Wuhan 430023, China
*   Correspondence: chenxl0811@whpu.edu.cn; Tel.: +86-188-5109-5957
†   These authors contributed equally to this work.

**Abstract:** *Cyclocarya paliurus* (Batal.) lljinskaja, the sole and multi-functional tree species of the family Juglandaceae, grows extensively in subtropical areas of China. Species-specific triterpenoids in its leaves have validated health-promoting effects, including hypoglycemic and hypolipidemic activities. To illustrate the effect of MeJA treatment on the accumulation and biosynthesis of triterpenoids in *C. paliurus* leaves at different stages of maturity, the contents of total triterpenoids and six triterpene compounds, along with the relative expression of three key genes, were detected. The results showed that the contents of triterpenes and expression patterns of the genes significantly differed among the samples. Different treatment times also had diverse effects on triterpenoid accumulation and gene expression. MeJA treatment had positive effects on total triterpenoids, cyclocaric acid B, and cyclocarioside A/B, especially in young leaves. Gene expression was highest in young leaves after 10 days of treatment, indicating that they were the most sensitive to MeJA. This study provided a reference for improving the accumulation of triterpenoids in *C. paliurus* plantations in the future.

**Keywords:** *Cyclocarya paliurus*; triterpenoids; triterpene compounds; expression; MeJA

## 1. Introduction

*Cyclocarya paliurus* is the only species in the family Juglandaceae and has medicinal, healthcare, material, and ornamental values. It is also called "Qing-qian-liu" or "Yao-qian-shu" because its fruit cluster is seen as a string of old Chinese copper coins [1–3]. Various bioactive constituents are found in the leaves of *C. paliurus*, including flavonoids, triterpenes, polysaccharides, and steroids, which have been shown to have pharmaceutical bioactivities [4–6]. Triterpenes, among the most important secondary metabolites in *C. paliurus*, attract much attention for their anti-diabetic, anti-inflammatory, and anti-cancer medicinal values [7,8]. Several species-specific triterpenoids, such as cyclocariosides A and I, have been extracted from *C. paliurus*. These compounds are remarkably sweet, with cyclocarioside A and cyclocarioside I being 200 and 250 times sweeter than sugar, respectively [9]. Therefore, they hold great potential as naturally occurring sweeteners, and it is important to demonstrate the molecular mechanism of triterpene biosynthesis and accumulation in *C. paliurus*.

Elucidating the biosynthetic pathways of secondary metabolites in medicinal plants is the basis for metabolic regulation and artificial synthesis of active ingredients. Triterpenoids, as typical secondary metabolites, occur at relatively low levels under natural conditions in plants; therefore, identifying the key enzyme genes involved in triterpene biosynthesis and the corresponding regulatory factors is necessary to improve their contents in *C. paliurus*.

Previously, three related key genes were identified and cloned in *C. paliurus*; their expression has been proven to affect triterpene accumulation significantly [10]. The key enzyme genes 3-hydroxy-3-methylglutaryl-CoA reductase *(HMGR)* and 1-deoxy-D-xylulose 5-phosphate reductase *(DXR)* are involved in the two upstream pathways (MVA and MEP) of triterpenoid biosynthesis, respectively. They synthesize substances with five carbon atoms, IPP (isopentenyl pyrophosphate, C5 unit) and DMAPP (dimethylallyl pyrophosphate, C5 unit), which form the basic skeleton of all terpenoids. With the increase in C5 units, branches leading to the production of monoterpenes, sesquiterpenes, diterpenes, and triterpenes appear in the biosynthetic pathway. The third key gene, *SQS* (*Squalene synthase*), is the first key enzyme gene involved in the triterpene synthesis branch of the terpenoid biosynthetic pathway.

Key enzyme genes for triterpene biosynthesis and regulatory factors play equally important roles in increasing the triterpenoid content of *C. paliurus*. Usually, chemical induction is used to increase the yield of secondary metabolites. Methyl jasmonate (MeJA) is one of the most commonly used exogenous chemical inducers and is used to increase the contents of triterpenoids in plants and medicinal fungi. Liu et al. treated *Sanghuangporus baumii* with several concentrations of MeJA. The accumulation of triterpenoids and expression of key genes involved in triterpenoid biosynthesis increased with increasing MeJA concentration, while the biosynthesis of other terpenoids was inhibited [11]. Key genes [3-hydroxy-3-methylglutaryl-CoA reductase (*HMGR*) and squalene epoxidase (*SE*)] and transcription factors [basic Helix-loop-helix (bHLH), MYB, and GRAS] involved in protostane triterpene biosynthesis in *Alisma orientale* were significantly enriched in a MeJA-treated group [12]. MeJA, as an important signal molecule in plant cells, can stimulate the expression of key genes related to triterpenoid biosynthesis in other plants, such as the expression of *SQS* in *Poria cocos* [13] and *CYP450* genes in *Panax ginseng* [14]. Based on these results, MeJA may also have a positive effect on the accumulation of triterpenes in *C. paliurus*.

The purpose of this study was to investigate the effects of MeJA treatment on the contents of total triterpenoids and six triterpene compounds in *C. paliurus* leaves at various stages of maturity. We also used quantitative real-time (qRT) PCR assays to detect the relative expression of key genes in the triterpenoid biosynthesis pathway. The results provided new insights into the metabolic regulatory network associated with triterpenoid biosynthesis in *C. paliurus* and provided suggestions for increasing the accumulation of triterpenoids in these trees.

## 2. Materials and Methods

### 2.1. Plant Materials

The clones used in the experiment were all grafted and had the same mother plant as the scion. The seed source was of Wufeng provenance. Leaflets were collected from different parts (Figure 1) of *C. paliurus* clone seedlings grown in a greenhouse [25/20 °C (day/night temperature) and 85% relative humidity] under long-day conditions (16-h day/8-h night) and a light intensity of 6000 lx. Upper leaves (U) were not fully unfolded, middle leaves (M) were just unfolded with a tender green color, and lower leaves (L) were fully unfolded and of a deep green color. Two groups of each sample were established; one group was used to detect triterpenoids and the other to determine relative gene expression.

### 2.2. Methods

Twelve healthy 2-year-old *C. paliurus* clone seedlings with good growth were selected for transplantation into a climate-controlled room 20 days before MeJA treatment. Half of the seedlings were placed in the experimental group and half in the control group. The experimental group was treated with 0.2 mM MeJA (pH 7.0, containing 0.01% Tween-20), and the control group was treated with water (pH 7.0, containing 0.01% Tween-20). The experiment lasted for 30 days, during which the seedlings were sprayed every 2 days. After the initial application of MeJA, leaf samples (U, M, and L) were collected on days 10,

20, and 30, respectively. Samples collected for total triterpene and triterpene compound determination were dried at 60 °C, and samples collected to investigate gene expression were stored at −80 °C.

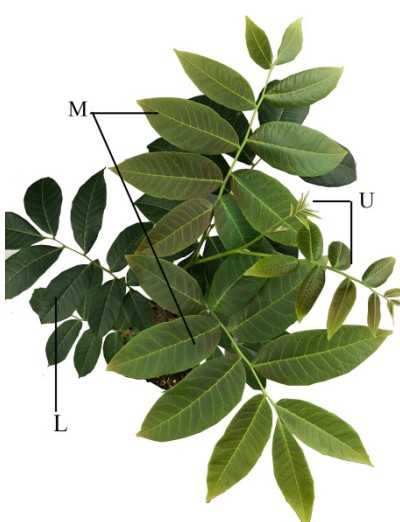

**Figure 1.** Top view of *C. paliurus*. Note: U, Upper; M, Middle; L, Lower.

### 2.3. RNA Isolation and cDNA Synthesis

A Plant RNA Kit R6827-01 (OMEGA, Bozeman, MT, USA) was used to extract and purify the total RNA from the leaf samples. A PrimeScript™ II First Strand cDNA Synthesis Kit (Takara, Dalian, China) was used to transform the RNA into cDNA.

### 2.4. Quantitative Real-Time PCR

All reactions were carried out on the StepOne Real-Time PCR System (Applied Biological Systems Company, Foster City, CA, USA) using a SYBR Green Real-Time PCR Master Mix Kit (Toyobo, Osaka, Japan) according to the manufacturer's instructions. Specific primers for target genes (*CpHMGR*, *CpDXR*, and *CpSQS*) were designed using Primer Premier 5.0 software (Table 1). Using 18S ribosomal RNA as the endogenous reference gene, relative gene expression was calculated using the $2^{-\Delta\Delta CT}$ method [15]. The PCR solution consisted of 2 μL template cDNA, 1 μL of each primer (10 μM) in the pair, 10 μL SYBR Green Real-Time PCR Master Mix, and 6 μL ddH$_2$O. The reaction conditions started at 95 °C for 30 s, and then entered 40 cycles of denaturation at 94 °C for 5 s, annealing at 55 °C for 35 s, and extension at 72 °C for 30 s.

**Table 1.** Sequences of primers used for qRT-PCR.

| Primer Name | Oligonucleotide Sequence (5′–3′) |
| --- | --- |
| *CpHMGR* F D | TTTAGCGATGGACATGAGCA |
| *CpHMGR* R D | GGAGTGGCAGAGCGTCAGAGGC |
| *CpSQS* F D | GAACAGGCTGGATGCGATAC |
| *CpSQS* R D | TCAATTATTTGGTCGTTTGG |
| *CpDXR* F D | GCTGGTTCAATGTAACTCTTC |
| *CpDXR* R D | CTCTATGACTCCTTGCTCCC |
| *18s* F | AGTATGGTCGCAAGGCTGAAA |
| *18s* R | CAGACAAATCGCTCCACCAA |

### 2.5. Determination of Triterpenoids

*C. paliurus* leaf tissue was ground into a powder and a 0.500 g sample was degreased with petroleum ether at 80 °C twice. Next, the residue was soaked for 12 h in 10 mL of ethanol (80%) and ultrasonically treated at 70 °C and 59 kHz for 40 min, before the extract was filtered through a 0.45 μm microporous membrane. The extracts were combined

and adjusted to a final volume of 10 mL with 100% ethanol. The determination of total triterpenoids was conducted using a colorimetric method as described by Chen et al. [16]. Detection was conducted at a wavelength of 550 nm. The total triterpenoid content was calculated according to the standard oleanolic acid curve and expressed as milligrams of oleanolic acid equivalent per gram of dry weight (mg·g$^{-1}$).

The contents of six triterpene compounds were analyzed using high-performance liquid chromatography (HPLC) and the results are summarized in Table 2 [17]. Prior to HPLC analysis (Agilent 1200 series HPLC system, Waldbronn, Germany), the extract was passed through a 0.4 µm polytetrafluoroethylene filter. A Waters 2489 ultraviolet detector (Waters, Milford, MA, USA) and an X-Bridge C18 column (250 × 4.6 mm) was used. The methods were as previously described by Sun et al. [18].

**Table 2.** Identification of six compounds from leaves of *C. paliurus* by developed HPLC–Q–TOF–MS.

| Compound | $t_R$ (min) | $[M - H]^-$ | MS/MS Fragment Ion (*m/z*) | Formula | Regressive Equation [a] |
|---|---|---|---|---|---|
| Arjunolic acid | 47.1 | 487.3429 | 445.2942; 401.3056; 389.2698 | $C_{30}H_{48}O_5$ | y = 5465.1x + 33,575 |
| Cyclocaric acid B | 49.7 | 485.3275 | / | $C_{30}H_{46}O_5$ | y = 8579.6x − 91,948 |
| Pterocaryoside B | 54.2 | 621.4001 | 521.3107; 489.3571 | $C_{35}H_{58}O_9$ | y = 3716.1x + 52,647 |
| Pterocaryoside A | 60.0 | 635.4162 | 535.3265; 489.3573 | $C_{36}H_{60}O_9$ | y = 3746.1x + 56,466 |
| Hederagenin | 60.5 | 471.3481 | 145.0286 | $C_{30}H_{48}O_4$ | y = 6054.7x + 25,521 |
| Oleanolic acid | 83.5 | 455.3549 | / | $C_{30}H_{48}O_3$ | y = 7144.8x + 963.97 |

[a] y is the peak area, while x is the concentration of each analyte (µg/mL).

### 2.6. Statistical Analysis

Differences in the total triterpenoid contents and those of six triterpene compounds, as well as gene expression, were analyzed using Duncan's test. Data were presented as mean ± standard deviation, and all statistical analyses were carried out with 95% confidence. SPSS 20.0 software (SPSS Inc., Chicago, IL, USA) was used for calculations and to conduct a Pearson correlation analysis.

## 3. Results

### 3.1. Effects of MeJA Treatment on the Contents of Total Triterpenoids in Leaves of C. paliurus

The total triterpenoid content of the *C. paliurus* leaf samples exhibited a significant difference between the experimental and control groups ($p < 0.05$). The triterpene contents of the upper leaves increased markedly after MeJA treatment (Figure 2), especially at 20 days. Compared with the control, the contents of triterpenes in the middle leaves treated with MeJA were significantly higher but showed the opposite trend at 30 days. In addition, the triterpene contents of the two groups showed no significant difference from those of the lower leaves at 20 days after treatment with MeJA. At 30 days after treatment, the triterpene contents of the experimental group (24.31 mg·g$^{-1}$) showed a slight decline compared to the control group (24.95 mg·g$^{-1}$).

### 3.2. Effects of MeJA Treatment on the Contents of Triterpene Compounds in Leaves of C. paliurus

Overall, MeJA treatment significantly affected the contents of triterpenoid compounds in the upper, middle, and lower *C. paliurus* leaves. Moreover, various triterpenes exhibited distinct differences in their contents (Figure 3). Among them, MeJA treatment significantly affected the contents of arjunolic acid and cyclocaric acid B in lower leaves. The arjunolic acid (Figure 3A) contents of upper, middle, and lower leaves after the same treatment period were also significantly divergent. In the experimental group, the content of cyclocaric acid B (Figure 3B) in the upper leaves was lower than that in the middle and lower leaves, but there was no difference between the 20- and 30-day treatment groups. Furthermore, the pterocaryoside B content after MeJA treatment was significantly higher than that in the control (Figure 3C), especially in the upper leaves. The same trend occurred in the 20- and 30-day treatment groups. The pterocaryoside A (Figure 3D) content in the upper leaves decreased considerably after 30 days of MeJA treatment, while the content in the middle

and, in particular, the lower leaves increased significantly. MeJA treatment for 20 days had an insignificant effect on the hederagenin (Figure 3E) content in the upper leaves, while the content in the middle leaves decreased significantly at 30 days. However, a promoting effect was seen overall. In addition, the oleanolic acid (Figure 3F) content in the upper and middle leaves, which had a higher initial content, decreased after MeJA treatment. In contrast, oleanolic acid content in the lower leaves was 15.36-fold and 1.76-fold higher than the control (CK) at Days 20 and 30 following MeJA treatment.

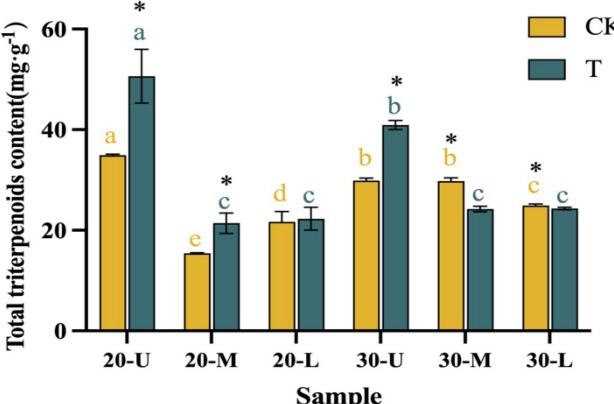

**Figure 2.** Effects of MeJA treatment on the content of total triterpenoids in leaves. Note: The experimental group was treated with 0.2 mM MeJA (T), and the control (CK) group was treated with water. The lowercase letters of the same color indicate statistical differences among different samples in the same group (test and control). * Significant differences between the same samples in separate groups. Letters that are the same indicate no significant difference according to Duncan's test ($p < 0.05$). The bars indicate the standard deviation for the three replicates. U, upper; M, middle; L, lower leaves; 20-, 20 days after MeJA treatment; 30-, 30 days after MeJA treatment.

### 3.3. Relative Expression of Key Enzyme Genes in Leaves at Different Stages of Maturity under Treatment of MeJA

In contrast to the content assay, additional upper leaves under treatment with MeJA over 10 days were sampled to detect gene (*CpHMGR*, *CpDXR*, and *CpSQS*) expression (Figure 4). Overall, the relative expression of genes involved in the biosynthesis of triterpenes was significantly increased by the MeJA treatment, particularly in the upper leaves; after 10 days of MeJA treatment, the gene expression in the upper leaves was 20 or even 1000 times higher than that in the control group, indicating that young leaves are more susceptible to MeJA. The expression of *CpHMGR* decreased with the extension of treatment time and was maintained at a relatively stable level after 30 days of treatment, while the expression of *CpDXR* and *CpSQS* showed the opposite trend.

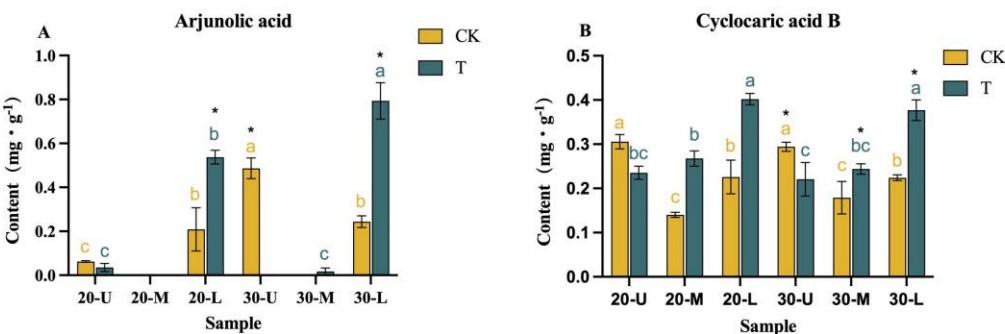

**Figure 3.** *Cont*.

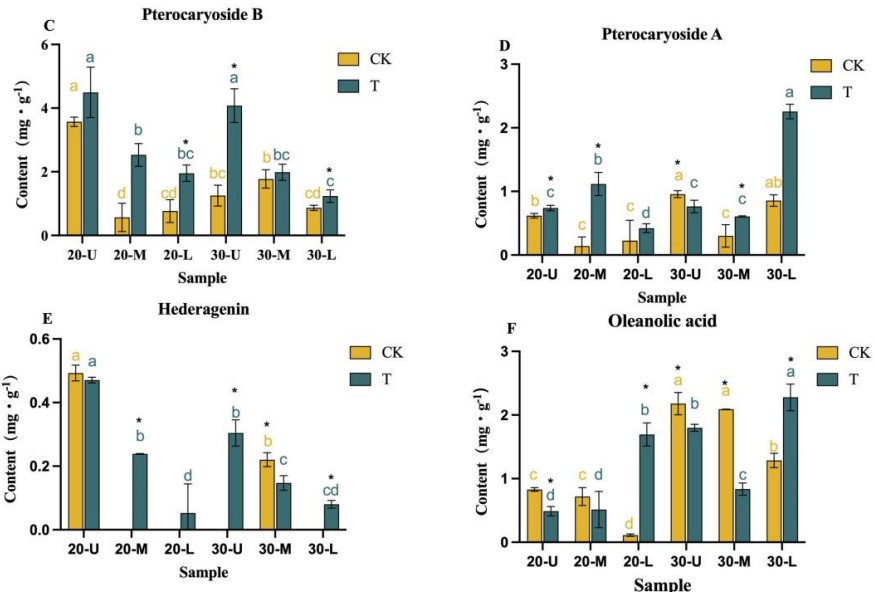

**Figure 3.** Effects of MeJA treatment on the contents of triterpene compounds in leaves of *C. paliurus*. (**A**) Arjunolic acid content in leaves; (**B**) Cyclocaric acid B content in leaves; (**C**) Pterocaryoside B content in leaves; (**D**) Pterocaryoside A content in leaves; (**E**) Hederagenin content in leaves; (**F**) Oleanolic acid content in leaves. Note: Lowercase letters of the same color indicate statistical differences among different samples in the same group (test and control). * Significant differences between the same samples in separate groups. Letters that are the same indicate no significant difference according to Duncan's test ($p < 0.05$). The bars indicate the standard deviation for the three replicates. U, upper; M, middle; L, lower leaves; 20-, 20 days after MeJA treatment; 30-, 30 days after MeJA treatment.

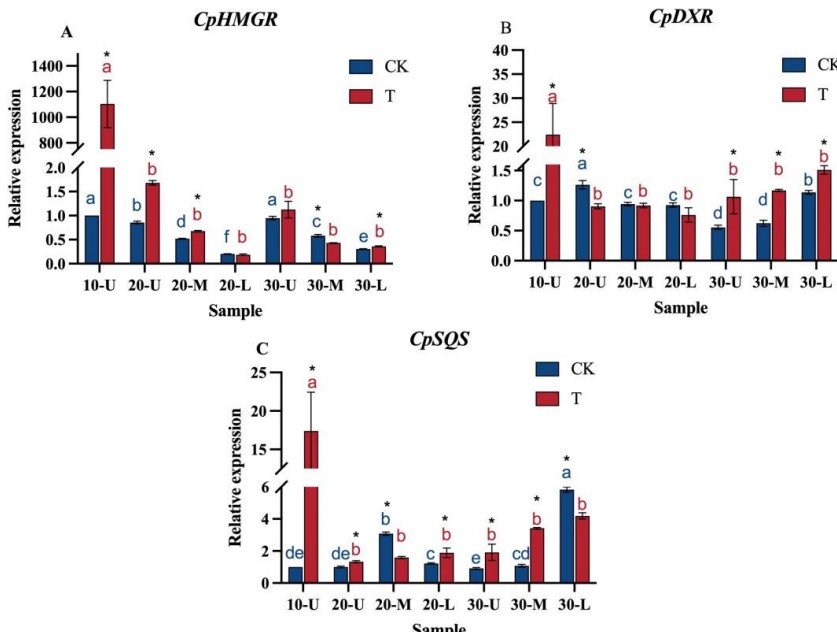

**Figure 4.** Relative expression of *CpHMGR*, *CpDXR*, and *CpSQS* in leaves of *C. paliurus* under treatment with MeJA. (**A**) Expression of *CpHMGR* in leaves; (**B**) Expression of *CpDXR* in leaves; (**C**) Expression of *CpSQS* in leaves. The same color of lowercase letters indicates statistical differences among different samples in the same group (test and control). * Significant differences between the same samples in separate groups. Letters that are the same indicate no significant difference according to Duncan's test ($p < 0.05$). The bars indicate the standard deviation for the three replicates. U, upper; M, middle; L, lower leaves; 20-, 20 days after MeJA treatment; 30-, 30 days after MeJA treatment.

## 4. Discussion

Several studies have demonstrated that triterpenoids in *C. paliurus* are key bioactive substances with pharmacological properties, and certain triterpenoids have previously been isolated from this species [19,20]. Exogenous hormone treatment to promote the accumulation of plant secondary metabolites has been applied to various plant species, such as *Artemisia carvifolia*, *Mentha canadensis*, and *Pinus massoniana* [21,22]. Jasmonates are generally considered to be effective stimulatory signal molecules. These occur as jasmonic acid and MeJA, which are formed after methylation [23–25]. MeJA regulates plant growth and development [26,27]. When plants are subjected to various stresses (drought, pathogenic bacteria, insect pests, and high salt), the expression of corresponding resistance genes can be induced, thus enhancing tolerance to these stresses. Studies have shown that the biosynthesis of triterpenoids in plants can be affected if jasmonic acid is applied as an exogenous hormone or if the content of jasmonic acid changes in response to environmental stress. Secondary metabolites such as flavonoids and triterpenes are also typical compounds produced as part of the plant defense response to various stresses. Many studies have been conducted on enhancing plant terpenoids through treatment with MeJA. This study analyzed the accumulation patterns of total triterpenes and six triterpene compounds, along with the expression patterns of three key enzyme genes (*CpHMGR*, *CpDXR*, and *CpSQS*) in *C. paliurus* leaves at different stages of maturity under MeJA treatment. The results provided a theoretical foundation for the targeted cultivation of *C. paliurus*.

Yang et al. reported that the content of saikosaponin showed an increase only in samples taken after the upregulation of gene expression. This discovery indicated that there was a time delay between gene expression and final product synthesis [28]. In our study, the greatest difference in triterpene content between the treatment and control groups appeared in later samples, while gene expression showed a significant difference in early samples. This indicated a lag between triterpene accumulation and the expression of relative genes. Moreover, the accumulation of triterpenoids and the expression of related genes differed in the samples taken at different sites and according to the duration of MeJA treatment. This confirmed that the downstream pathways of triterpenoid biosynthesis were complex. Changes in external factors might affect one or several pathways corresponding to one or several products. This difference has also been shown in various other species. Qi et al. analyzed the transcriptome data of MeJA-treated *M. canadensis*, and the results showed that differentially expressed genes were significantly enriched in the monoterpene synthesis pathway. In addition, Zeng et al. screened 11 *CYP450* genes possibly involved in ginsenoside biosynthesis through transcriptome data analysis and analyzed the expression patterns of relative genes under treatment with MeJA using qRT-PCR. The results showed that MeJA strongly induced the expression of partial *CYP450* genes, and different durations of MeJA treatment corresponded to different transcription levels. Moreover, Yao et al. reported that MeJA treatment increased the contents of terpenoids in *P. massoniana* needles, especially the contents of monoterpenoids and diterpenes. Jasmonic acid is also an effective promoter in the biosynthesis of sesquiterpene in *A. carvifolia*, and its mechanism of action is slowly being revealed.

Under MeJA induction, the key enzyme genes for triterpenoid biosynthesis can also be upregulated by their promoters. In *Panax quinquefolium*, treatment with 0.25 mmol/L MeJA can increase the induction rate of *SE* gene promoters by up to 2435.4-fold, which affects the accumulation of final products [29]. Like jasmonic acid, regulating the expression of transcription factors (WRKY, MYB, and MYC2, etc.) is one of the important ways MeJA induces triterpenoid biosynthesis. Based on previous studies, MeJA can stimulate relative transcription factors and promoters of key genes so as to upregulate the expression of key enzyme genes and increase the accumulation of triterpenoids. To date, this has been studied in *Conyza blinii H. Lév.* [30], *P. ginseng* [31], *Withania somnifera* [32], etc. In *C. paliurus*, MeJA treatment showed a tendency to promote triterpene accumulation. The total triterpene content was most substantially promoted in the upper leaves, while the effect on the

accumulation of each triterpene compound differed. MeJA treatment greatly induced the contents of cyclocaric acid B and pterocaryoside B in the leaves of *C. paliurus*, indicating that MeJA only affected certain pathways. Since the downstream pathways of most of the unique triterpenoids in *C. paliurus* are unclear, it is impossible to determine which pathway is stimulated according to the change in the triterpenoid contents. The results showed that the increase in pterocaryoside B content was the most significant, whether treatment was over 20 or 30 days. Secondly, the content of total triterpenes was significantly increased in the young leaves (upper), and the expression level of each gene was also dozens or even thousands of times higher than that in the middle and lower leaves, indicating that the young leaves were the most sensitive to stimulation. Continued spraying with an appropriate amount of MeJA before picking young leaves will improve product quality and provide more economic benefits.

The results of the correlation analysis of the total triterpenoid content, six triterpenoid compounds, and the expression of three key enzyme genes in leaves from different parts of *C. paliurus* are shown in Figure 5. The correlation between gene expression and the content of various triterpenes showed diversity. In this study, the contents of total triterpenoids were positively correlated with the expression of *CpHMGR*, indicating that treatment with MeJA promoted the MVA pathway in triterpene synthesis. The expression of *CpHMGR* and *CpDXR* were positively correlated with the contents of different triterpene compounds, indicating that the biosynthesis of different triterpene compounds was different in the upstream pathway. Chen et al. [10] explored the relationship between gene expression (*HMGR, DXR, SQS*) and triterpene content in leaf and shoot, respectively, during the growing season in *C. paliurus*; however, no significant correlation was shown between *CpHMGR* expression and total triterpenoid content, which is inconsistent with our findings. The main reason could be that different methods of sampling (sampling time and sampling part) produce varied results.

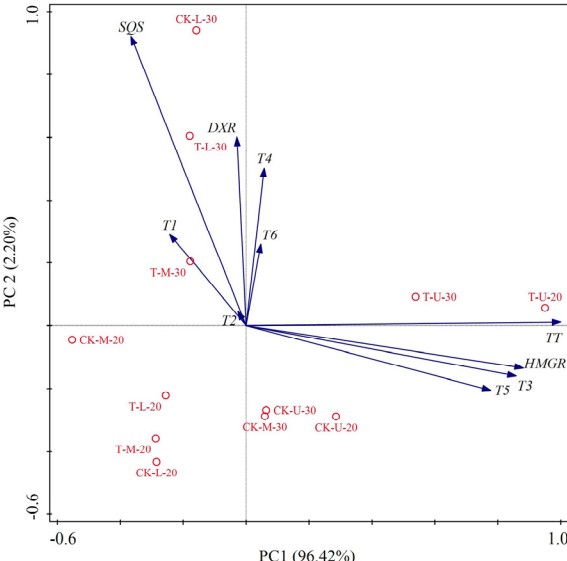

**Figure 5.** Principal component analysis score plot and loading plot describing the relationship among triterpene content and key enzyme gene expression. U, Upper; M, Middle; L, Lower; 20-, 20 days after MeJA treatment; 30-, 30 days after MeJA treatment; T, MeJA treatment group; CK, blank control group; TT, total triterpenoids; T1, arjunolic acid; T2, cyclocaric acid B; T3, pterocaryoside B; T4, pterocaryoside A; T5, hederagenin; T6, oleanolic acid.

Enzymes are involved in the biosynthesis of various types of terpenoids, such as monoterpenoids, sesquiterpenoids, diterpenoids, and sterols. They are also involved in the production of other primary/secondary metabolites and are not exclusively responsible for triterpene biosynthesis. Furthermore, it is likely that most of the intermediates catalyzed

by *CpHMGR* and *CpDXR* in young shoots and roots are used for growth and flowering, transported to the leaves, or even utilized in the biosynthesis of other compounds. These results indicated a complex relationship between upstream gene expression and triterpenoid accumulation. In addition to the complex branches of the downstream gene synthesis pathways, the transfer of intermediates, and synthesis of end-products also affect the process. Previous studies have confirmed that the expression of *CpSQS* in leaves is the highest, indicating that it may accept intermediate products from other tissues and organs and then combine them with the intermediate products of its synthesis to complete the synthesis of the final triterpenoids.

## 5. Conclusions

MeJA, as a chemical inducer, effectively promoted the accumulation of triterpenes in *C. paliurus*, exhibiting significant promotion of cyclocaric acid B and cyclocarioside A/B production. In addition, MeJA had a significant regulatory effect on the expression of *CpSQS*. The effect of treatment with MeJA on *C. paliurus* was of great significance to the future application of multi-omics analysis techniques for exploring the biosynthetic pathways of these compounds and elucidating the function of *CpSQS*.

**Author Contributions:** Conceptualization, Q.X. and Z.W.; methodology, Q.X., Z.W. and X.C.; software, Q.X.; validation, Z.W.; formal analysis, Q.X. and Z.W.; investigation, Q.X.; resources, X.C., S.Z. and S.C.; writing—original draft preparation, Q.X. and Z.W.; writing—review and editing, X.C.; visualization, X.C. and X.D.; supervision, X.D.; funding acquisition, X.C., X.D., S.Z. and S.C., Q.X. and Z.W. contributed equally to this paper. All authors have read and agreed to the published version of the manuscript.

**Funding:** This study was supported by the National Natural Science Foundation for Young Scholars of China (32201604), Research and Innovation Initiatives of WHPU (2022Y41), the Doctoral Research Funding Project of Wuhan Polytechnic University (2021RZ070). The authors also thank the financial support from the Hubei Provincial Natural Science Foundation Project (2022CFB945) and the Educational Commission of Hubei Province of China (Q20211612).

**Data Availability Statement:** The data presented in this study are available on request from the corresponding author.

**Conflicts of Interest:** The authors declare no conflict of interest.

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
