# Peer review of "Effects on the Synthesis and Accumulation of Triterpenes in Leaves of Cyclocarya paliurus under MeJA Treatment"

_forests, doi:10.3390/f14091735_

Round 1
Reviewer 1 Report
Abstract: The last sentence should be fully expand, i.e., what are the proposed biosynthesis pathway?
Introduction:
L38-52: Describe specific triterpenoids in this plant, and the medicinal values.
How would these plant contribute as part of the forest?
Material and method
2.1 Explain how seedling was obtained?
2.2 the problem of seeding for this experiment is that how do we prove that all seedling are of the same clone? describes terms L, M, U in the body i.e., the position of leaf pair.
2.3 Abbreviation should be fully expanded.
2.5 Describe 6 triterpene monomers used for HPLC, and how do you calculate the contents? Extrapolate from the standards? What are also the standards used?
Results:
Figure 2 describe CK and T
The result in Figure 2 and the combined contents of the triterpenoids in Figure does not seem to be in alignment. I seemed that only 2 compounds contributed to the overall contents of triterpenoids.
I would suggest to use multi-variable tools such as PCA, or PLS to analyse the correlation, which giving the better picture how genes corresponded to the biosynthesis of each types of triterpenes.
Line271-291: What would be the scenario of triterpenoid biosysthesis describe from the experiment?
I expect to see the conclusion of the finding in the last session too.
Reviewer 2 Report
The manuscript contains some interesting data but it is not well prepared, badly written and it contains many serious mistakes.
1. There is a chaos in using “tritepene” and “triterpenes/triterpenoids” in the text. Sometimes Authors wrongly use “riterpene” in singular, whereas it should be used in plural (triterpenes):
Line 32. triterpene (triterpenoids) –please correct triterpene to triterpenes.
Line 34. As a name of a class of compounds, please correct triterpene to triterpenes.
In turn, the word “triterpene” can be used as adjective, e.g., triterpene biosynthesis, triterpene accumulation, triterpene compounds etc., such expressions are correct.
2. The great error in terminology is the term “triterpene monomer”. Monomer – by definition –is a molecule that can be bonded to other identical molecules to form a polymer. In the case of terpenoids it is IPP, and its isomer DMAPP. The Authors applied the term “monomer” to typical triterpenoid compounds, e.g., occurring widely in higher plants C30 compound oleanolic acid. Oleanolic acid is not a monomer, it is not forming any polymers with other molecules of oleanolic acid.. Neither other listed compounds.
This mistake in terminology is striking, because there are numerous publications that properly described these compounds (e.g., Sun et al., Front. Plant Sci. 2022, 13: “Previous studies have analyzed the composition of compounds in leaves of C. paliurus, including six main triterpene compounds (e.g., arjunolic acid, cyclocaric acid B, pterocaryoside B, pterocaryoside A, hederagenin, and oleanolic acid”).
This mistake should be carefully corrected in all the text of the manuscript.
3. Another error in terminology is misunderstanding with the term “saponins” (Line 141). Saponins are not just triterpenoids, they are triterpenoid (or steroidal) glycosides. Why suddenly saponins? Indeed, pterocaryoside B and pterocaryoside A contain sugar in the molecule, but the other four triterpenoids does not. So, finally, what was determined, triterpenoids or saponins? Please clarify it in the text.
4. Another error – semiterpenoids (line 280). Such compounds does not exist. Sesquiterpenoids?
5. Some sentences are awkward, e.g., (line 40 “Triterpenoids, as typical secondary metabolites, are characterized as having low content under natural conditions in plants”. it is better to write “Triterpenoids, as typical secondary metabolites, occur in plants in relatively low content under natural conditions”…The whole text should be carefully read and corrected to improve the style of writing. For example: (lines 172-174): “However, it still showed a promoting effect on the whole.” (what means :the whole”?). “In contrast, the lower leaves with the lowest initial content reach to high value.” (such sentences should be written more clearly, e.g., the content of oleanolic acid in lower leaves was initially the lowest, and afterwards it reached a high value” specify how high? e.g. 3-fold?)
6. Some parts of method description are not understandable:
Line 91. “and the control group was with water rather than MeJA: (why “rather than MeJA”? Please delete this fragment)
Line 93. “The seedlings were sprayed every two days and last for 30 days. Leaves in the upper (U), middle (M) and lower (L) were collected after 20 days and 30 days of the treatment respectively.” What it means? The scheme of the experiment is not clear. That the seedlings were sprayed every two days during 30 days? And these 20 and 30 days of the treatment – was it 20 and 30 days AFTER the last spraying? Or just after the last spraying (so in the case of 20 days – still during the experiment? I guess no). What means “respectively” – it is three types of the leaves and two time points? “Leaves in the upper (U), middle (M) and lower (L)” – maybe Upper (U), middle (M) and lower (L) leaves, otherwise it is not grammatically correct.
Line 95. “Samples of total triterpenes and triterpene monomer determination were dried …(…), and the other part for expression were stored at…”. Please correct to: “Samples collected for total triterpenes and triterpene monomer determination were dried…(..), and samples collected for investigation of gene expression were stored at …”.
7. Please correct all figure captions (missing big letters, dots etc.).
The quality of English language is low, some sentences are awkward, not written in scientific style.
Round 2
Reviewer 1 Report
I am still not sure if the term seedling is correct, may be clone? The term "seedling" is used to refer to a young plant that has grown from seed. Seedlings are not clones of their parents, as they inherit genes from both parents.
Point 6: Please add the table in the MS.
n/a
Author Response
Thank you for your comments concerning our manuscript entitled "Effects on the synthesis and accumulation of triterpenes in leaves of Cyclocarya paliurus under MeJA treatment" (forests-2517328). Those comments are valuable and very helpful for revising and improving our work and providing constructive suggestions for our research.
Response to Reviewer 1 Comments
Point 1: I am still not sure if the term seedling is correct, may be clone? The term "seedling" is used to refer to a young plant that has grown from seed. Seedlings are not clones of their parents, as they inherit genes from both parents.
Response 1: Thank you for your comments. We rewrote the sentence, please see the latest MS. Line 82.
Point 2: Please add the table in the MS.
Response 2: Done!
Reviewer 2 Report
The Authors have corrected the mistakes, and improved the manuscript according to my suggestions.
Author Response
Thank you for your comments concerning our manuscript entitled "Effects on the synthesis and accumulation of triterpenes in leaves of Cyclocarya paliurus under MeJA treatment" (forests-2517328). Those comments are valuable and very helpful for revising and improving our work and providing constructive suggestions for our research.